# Disposable Potentiometric Sensory System for Skin Antioxidant Activity Evaluation

**DOI:** 10.3390/s19112586

**Published:** 2019-06-06

**Authors:** Khiena Brainina, Aleksey Tarasov, Ekaterina Khamzina, Yan Kazakov, Natalia Stozhko

**Affiliations:** 1Ural State University of Economics, 62/45, 8 Marta/Narodnoi Voli St., 620144 Yekaterinburg, Russia; tarasov_a.v@bk.ru (A.T.); xei260296@mail.ru (E.K.); yankaz@yandex.ru (Y.K.); sny@usue.ru (N.S.); 2Ural Federal University named after the first President of Russia B. N. Yeltsin, 28, Mira St., 620078 Yekaterinburg, Russia

**Keywords:** potentiometric sensory system, contact hybrid potentiometric method, screen-printed electrodes, antioxidant activity of skin

## Abstract

The skin is a natural barrier between the external and internal environment. Its protective functions and the relationship of its state with the state of health of the organism as a whole are very important. It is known that oxidant stress (OS) is a common indicator of health status. This paper describes a new sensory system for monitoring OS of the skin using antioxidant activity (AOA) as its criteria. The contact hybrid potentiometric method (CHPM) and new electrochemical measuring scheme were used. A new sensory system, including disposable modified screen-printed carbon and silver electrodes covered by membrane impregnated by mediator, was developed. Its informative ability was demonstrated in the evaluation of the impact of fasting, consumption of food and food enriched by vitamins (antioxidants) on skin AOA. This device consisting of a sensory system and potentiometric analyzer can be used in on-site and in situ formats.

## 1. Introduction

The skin acts as a mechanical and chemical barrier separating the external world and the internal environment of the body; representing a biological boundary and biosurface on which complex biochemical processes occur. These processes are regulated under the influence of several external and internal factors and have different features at the surface and in deep layers of the skin; as additionally, the structure and properties of the skin layers differ [1,2,3]. The main effects of the external environment on the skin are expressed primarily in the action of ultraviolet radiation, constant contact with atmospheric oxygen, exposure to solid atmospheric microparticles, and are realized through oxidative stress reactions occurring in the skin, causing damage of the skin structural elements, accelerated skin aging, and skin diseases (including cancer) [4,5,6]. Internal exposure can also lead to the development of oxidative stress throughout the increased production of reactive oxygen species (ROS); this is associated with a violation of DNA synthesis and the accumulation of genetic changes in the skin cells, disruption of lipid synthesis in the epidermis, and damage of the collagen matrix [7,8]. Sources of ROS in the skin are: Mitochondrial electron transport chain, peroxisomal enzymes (associated with fatty acid oxidation and glyoxylate/dicarboxylate metabolism), and endoplasmic reticulum enzymes (from the cytochrome p450 system and protein disulfide isomerase in combination with the endoplasmic reticulum oxidoreductin-1), NADPH oxidases of the membranes and the cytosol, the Fenton and Haber–Weiss reactions, and enzymes such as cyclooxygenases and lipoxygenases (arachidonic acid metabolism), and xanthine oxidases (purine catabolism) [9,10].

Harmful influences on the skin are opposed by complex regulated protection systems such as the antioxidant system of the skin [11], consisting mainly of low-molecular antioxidants [12,13,14,15], and the system of chromophores—molecules (majority of them are melanin, hemoglobin, nucleic acids, bilirubin, and aromatic amino acids; e.g., phenylalanine, tyrosine, and urocanic acid) capable of absorbing the energy of sunlight in a wide range of wavelengths (250–3000 nm), and then dissipate it as heat or transfer it to a nearby molecule with the formation of photoproducts, including reactive oxygen species [16].

The activity of metabolic processes is maximal in the epidermis [10]; the content of enzyme and non-enzyme (low molecular weight) antioxidants, respectively, is much higher in the epidermis than in the derma [12,15,17]. In addition, the surface of the skin is able to emit low molecular weight antioxidants [1]. Skin lesions caused by the deficiency of various vitamins in food (peeling of the skin due to deficiency of vitamin A, pellagra due to deficiency of nicotinic acid (vitamin PP), halite used by the deficiency of vitamin B6 (pyridoxine)) are described. To date, many studies have been conducted to demonstrate the effect of food antioxidants and various food components on skin aging: The antiaging (anti-wrinkle) effect of vitamin C, vegetables, olive oil, legumes, etc., and the opposite effect of higher fat and carbohydrate intake [18]. The consumption of polyphenols-rich food can provide photoprotective effect against UV radiation and counteract against skin photocarcinogenesis [19]. Fasting effects on skin are not completely described: Thus, it has been shown that long term restriction in the appropriate foods leads to decrease of glycation products (carboxymethyl lysine and pentosidine) and age-related accumulation of these metabolites in skin collagen, but the authors postulate the strong need for evidence-based suggestions and guidelines concerning the assessment of fasting effects on the skin [20]. Future high-quality clinical trials are needed to establish the effects, indications, and dosages of antioxidants, vitamins, and nutrition supplements to reduce OS and prove any antiaging and cancer-prevention effects in the skin. 

The presented data serves as a justification for the determination of the antioxidant activity (AOA) of skin [21]. Thus, monitoring of AOA is an important task, as its solution is the possible key to the diagnosis of skin aging and skin diseases (including malignant processes). However, the features of the structure and function of the skin, the multifactorial influence on the skin’s redox reactions make this task very difficult. We believe that an integral approach to the assessment of oxidative stress using the value of the AOA as an estimation parameter of its severity level is preferable. 

As a rule, the concentration of antioxidants in the skin is measured by invasive methods based on the analyzing of skin tissue homogenates. Non-invasive methods are not widely used. These include, for example, the method of resonance Raman spectroscopy [22]. This method allows the determination of only skin carotenoids (β-carotene, lycopene), which make up only a small part of antioxidants. Other water-soluble and fat-soluble antioxidants remain outside the scope of the data obtained by those methods. A nanoparticle-based paper sensor for thiols evaluation in human skin was recently described [23], but it also solves only one particular task.

Another approach is described in a series of works [24,25,26]. Ron Cohen is apparently one of the first to have proposed a non-invasive method for determining skin AOA [27].

Later, we proposed a potentiometric method for non-invasive evaluation of skin AOA [28]. This hybrid potentiometric method, developed in the works [29,30], allows the estimation of the total AOA of the studied object. It is based on the interaction of the skin epidermis antioxidants with the mediator system introduced in electrically conductive gel [28] or membrane [30], which is placed on the skin. It is a one-step method in which stages of the extraction of determined substances from the skin and measurements are combined in time and space. The source of information on the total AOA of the skin epidermis is the shift of the electrode potential in the mediator as a result of mediator interaction with the antioxidants diffusing from the epidermis.

Goals of the work are: (i) To develop a new disposable sensory system to be used in the hybrid potentiometric analysis method; (ii) to use it as measuring means in contact with skin, and thus to develop a contact variant of the aforementioned method; (iii) to demonstrate its applicability for evaluation of the AOA of skin as a skin sensory system; and (iv) to demonstrate the informative ability of the proposed sensory system in the evaluation of the impact of fasting, and consumption of food and food enriched by vitamins (antioxidants) on skin antioxidant activity.

## 2. Materials and Methods

### 2.1. Instruments

To reach the goals of the study, the equipment mentioned below was used: The portable potentiometric analyzer PA-S (USUE, Yekaterinburg, Russia) for potentiometric studies; the voltammetric analyzer IVA-5 (IVA Ltd., Ekaterinburg, Russia) for the polarization of the silver screen-printed electrode in the modification process; the manual screen printing device SPR-10 (DDM Novastar Inc., Warminster, PA, USA) for the application of conductive carbon ink and silver paint on the substrate; the deionizer Akvalab-UVOI-MF-1812 (JSC RPC Mediana-filter, Moscow, Russia) for obtaining deionized water with a specific resistance of 18 MΩ cm); an 18–26 cm cuff complete with pediatric sphygmomanometer LD-80 (Little Doctor Int. (S) Pte. Ltd., Yishun, Singapore) to fix the sensory system on the wrist.

### 2.2. Chemicals and Reagents

The following reagents were used: K_3_[Fe(CN)_6_)], K_4_[Fe(CN)_6_] × 3H_2_O (AO Reachim Ltd., Moscow, Russia); NaCl (OJSC Mikhailovskiy Plant of Chemical Reagents, Russia); KCl, Na_2_HPO_4_ × 12H_2_O (CJSC Vekton, St. Petersburg, Russia); KH_2_PO_4_ (NevaReaktiv Ltd., St. Petersburg, Russia); sodium citrate Na_3_C_6_H_5_O_7_ (JSC ChemReactivSnab, Ufa, Russia). These reagents were chemically pure. Acetone was pure for analysis (Component-reaktiv Ltd., Moscow, Russia). Other chemicals were: 0.05 M HAuCl_4_ solution (RPE Tomanalyt Ltd., Tomsk, Russia); L-ascorbic acid BioXtra ≥99% (Sigma-Aldrich Co., St. Louis, MO, USA); Cementit universal (Merz + Benteli AG, Niederwangen, Switzerland) were used. Askorutin tablets containing ascorbic acid (vitamin C) 50 mg and rutoside (flavonoid from vitamin P group) 50 mg in each tablet was bought in an official pharmacy.

### 2.3. Materials

Glass fiber with a thickness of 0.35 mm STEFI mark (PJSC Elektroizolit, Khotkovo, Russia) was used as substrate for screen-printed electrodes. The membrane MFAS-OS-2 (CJSC STC Vladipor, Vladimir, Russia) based on cellulose acetate served as a mediator holder in the sensory system. Polyester mesh served as the stencil for applying of the electrodes. Carbon conductive ink (Guangzhou Print Area Co. Ltd., Guangzhou, China) and silver conductive paint (Mechanic products in Vietnam, Nationwide distribution system, Vietnam) were used.

### 2.4. Investigated Persons

All volunteers involved in the study provided informed consent. The study was conducted in accordance with the Declaration of Helsinki, and the protocol of the study was approved by the Ethics Committee of “Medical Technologies” JSC (Project identification code 16-01-18 MT-AO), in accordance with the rules of Good Clinical Practice.

A group of healthy volunteers aged 18–34 years old (n = 13) was investigated. Inclusion criteria: Healthy volunteers with skin type 1–2 (according to skin phototypes by Fitzpatrick [31]). Exclusion criteria: Skin phototypes 3–6, skin diseases, known chronic diseases, smoking, visiting solarium, consumption of b-carotene and other antioxidant nutrients supplements, and use of cosmetic creams and oils less than 24 h before the procedure. Volunteers included in the study had no chronic diseases, their anthropometric data and basic indicators of objective examination (blood pressure, heart and respiratory rate, clinical blood and urine tests, electrocardiography and chest X-ray data) were within normal ranges, they had no vegetarian or other specific diet. Six volunteers took two Askorutin tablets, each containing ascorbic acid (vitamin C) 50 mg and rutoside (flavonoid from vitamin P group) 50 mg, after first the measurement, during breakfast.

### 2.5. Methods and Calculations

The electrochemical method used includes a chemical reaction (1) that serves as signal forming, and a electrochemical stage, that generates a signal. These processes occur simultaneously or sequentially after the sample is brought into contact with the sensory system.

The reaction Equation (1) describes interaction of antioxidants which diffuse from the object under study, with the oxidized form of the mediator (K_3_[Fe(CN)_6_]).
(1)$$a\times {\left[Fe{\left(CN\right)}_{6}\right]}^{3-}+\;b\times AO\;=\;a\times {\left[Fe{\left(CN\right)}_{6}\right]}^{4-}+\;b\times AO_{Ox},$$
where: AO—antioxidants, AO_Ox_—oxidized form of the antioxidants, *a* and *b*—stoichiometric coefficients.

The scheme of delivery of the signal-forming agent (the reduced form of the mediator, K_4_[Fe(CN)_6_]), which occurs as a product of the reaction (1) to the surface of the indicator electrode, is presented in Figure 1. As a result of changes in the ratio of oxidized and reduced forms of the mediator, a shift in the potential of the electrode contacting with the membrane is observed. This shift, recorded 10 min after starting the measurement, served as an analytical signal.

The potential shift is described by Equation (2):(2)$$\Delta E=b\times \mathrm{lg}\left(\frac{(C_{Ox}-X)\times C_{Red}}{(C_{Red}+X)\times C_{Ox}}\right),$$
(3)$$X=AOA=\frac{C_{Ox}-\alpha C_{Red}}{1+\alpha },$$
(4)$$\alpha =\frac{C_{Ox}}{C_{Red}}\times 10^{\frac{\Delta EnF}{2.3RT}},$$
where *C_Ox_* is K_3_[Fe(CN)_6_] concentration, M; *C_Red_* is K_4_[Fe(CN)_6_] concentration, M; *X* = AOA; b = 2.3 RT/nF; *R*—universal gas constant, J mol^−1^ K^−1^; *T*—temperature, K; *n* = 1—number of electrons involved in the process; *F*—Faraday constant, C mol^−1^; Δ*E*—potential shift, V.

All measurements were performed 1–6 times. Microsoft Excel 2010, with an accepted significance level α = 0.05, was used for data processing. The results are presented as X ± ΔX, where X is the mean value and ΔX is its standard deviation. The reproducibility of the analysis results was characterized by the value of the relative standard deviation (RSD), and the correctness of the analysis results—by the value of recovery, defined as the ratio of the found concentration of the model antioxidant (L-ascorbic acid) to the introduced quantity.

## 3. Sensory System Design

### 3.1. Fabrication of the Sensory System

The sensory system proposed consists of three parts: An indicator electrode (new), a reference electrode (new), and a special membrane, impregnated by mediator.

### 3.2. Gold Nanoparticles Suspension Synthesis

A gold suspension was synthesized by chemical reduction of aqueous solution of chloroauric acid (HAuCl_4_) by sodium citrate (Na_3_C_6_H_5_O_7_) in accordance with the standard Turkevich method [32]. The sol of gold nanoparticles having red color was obtained with the ratio HAuCl_4_:Na_3_C_6_H_5_O_7_ equal to 1:5 (M/M).

### 3.3. Modified Carbon Screen-Printed Electrode

The substrate was washed with acetone, ethanol, and deionized water, carbon conductive ink was applied to it and kept for 30 min in a drying chamber at a temperature of 110 °C. Individual electrodes had a size of 35 × 2 × 0.35 mm. The middle part, which separates the working and contact zones, was isolated with a mixture of Cementit and acetone in a ratio of 1:5 by volume so that the working area of the electrode was 8 mm^2^ (4 × 2 mm). A 10 µl suspension of gold nanoparticles was applied to the working area of the electrode, the electrode was dried at room temperature in the air. This modified electrode was further designated as CSPE/AuNPs and used as an indicator electrode of proposed sensor system.

### 3.4. Modified Silver Screen-Printed Electrode

The important part of the sensory system is the new reference electrode. Instead of the common silver chloride electrode, the potential of which depends on the concentration of the components of the mediator, a silver electrode covered with a mixed precipitate of silver chloride and ferricyanide is used as a reference electrode. The flat configuration of the new reference electrode allows the problem of standardization of the contact of the sensory system with the skin to be solved and to make the measurement process more convenient. As a result, as will be shown below, the reproducibility of results is improved.

The manufacturing technology of the reference electrode did not differ from that described above for the indicator electrode. The difference was in the way it was modified. In this case, a mixed precipitate of silver chloride and silver ferricyanide was formed on the electrode surface, in the process of electrode polarizing at potential 0.325 V vs. saturated silver/silver chloride electrode. This modified electrode was further designated as AgSPE/mod and was used as a reference electrode.

### 3.5. Membrane

The membrane served as a carrier of the mediator. To this end, it was kept for at least 3 min in a phosphate-buffered saline (PBS) containing 1 mM K_3_[Fe(CN)_6_] and 0.05 mM K_4_[Fe(CN)_6_] [30], before use, the excess solution allowed to drain off, and applied to electrodes (for model studies) or on the skin in the study of the latter.

### 3.6. Scheme and Measurement Technic

The measuring scheme used is shown in Figure 2. Such a circuit allows the correct results to be obtained, due to the fact that measurements are carried out at the same time in similar conditions. Another advantage of the measurement method proposed in this work is the reduction of time, as two results are obtained simultaneously in one measuring period. The reliability of the contact of the electrodes with the membrane and the object under study is ensured by placing a load of 70 g on the plate placed on the sensory system (when working with model systems) or pressing the sensory system to the skin with the help of cuff (pressure 35–40 mm Hg [30]).

Measurements are started after the sensory system is brought into contact with investigated sample or skin. Potential-time dependence is registered and processed with the use of potentiometric analyzers PA-S.

## 4. Results and Discussion

### 4.1. Study of the Kinetics of Stabilization and Reproducibility of the Potentials of the Electrodes Values

The scheme shown in Figure 2a was used. Three electrodes of the same type were switched in it, which made it possible to investigate 9 electrodes by conducting 3 measurements. The results characterizing the kinetics of the stabilization and reproducibility of the potentials of the electrodes values are presented in Table 1.

From Table 1 it can be seen that the CSPE is conceding in terms of the aforementioned characteristics to the CSPE/AuNPs electrode, therefore, the CSPE/AuNPs were used as the indicator electrode in the sensory system. The AgSPE/mod was used as a reference electrode in the sensory system, the potential of which is stabilized fairly quickly and is well reproduced.

### 4.2. Investigation of Sensory System in Model Conditions

L-ascorbic acid was used as the model antioxidant. The measuring scheme, shown in Figure 2, where E_1_ and E_3_ are CSPE/AuNPs, and E_2_ is AgSPE/mod. Measurements with E_1_ − E_3_ and E_2_ − E_3_ allow two independent results to be obtained simultaneously. Under these conditions, the potential of the sensory system was established within 200 s. The results are presented in Table 2.

As seen in Table 2, in the 30–900 μM-eq AOA interval the relative standard deviation does not exceed 8%, and recovery tends to 100%. This interval of the AOA model solutions correlates to the AOA of the skin of the respondents, previously obtained by another method [30].

### 4.3. Sensory System in AOA of Skin Evaluation

#### 4.3.1. Measurement of Skin AOA and L-Ascorbic Acid Recovery

Measurements were done as described in Section 4.2. Data are given in Table 3. Under these conditions, the potential stabilization was slower (600 s) than in the previous case, which is due to diffusion limitations.

The relative standard deviation in the evaluation of skin AOA is 9–20%, recovery of L-ascorbic acid introduced into the membrane varies between 100–110%. The data presented indicate that the proposed method and the sensory system provide results with good analytical characteristics.

#### 4.3.2. Practical Application of the Developed Sensory System

The results of the AOA of the skin determination of the 12 volunteers (six male and six female), aged 18–23 years during the fasting period in the early morning hours after waking up (duration of fasting approximately 1–1.5 h) and 100 ± 10 min after taking a regular meal (continental breakfast, without fruits and juices) are shown in Table 4. As a rule, after breakfast, an increase in the AOA of skin is observed, which indicates the presence of “fasting” (oxidative) stress (probably due to depletion of stocks of antioxidants), and its reduction after meals. The exception is when the initial (fasting) AOA exceeds the average value. Probably, the volunteers who are presented in the table under numbers 6 and 10–12 are initially more resistant to oxidative stress and fasting does not affect them as noticeably as other respondents. Regular physical activity (fitness) and special sports nutrition, too, possibly can influence the AOA of the skin, but a special study is needed to investigate this relationship. In our study, there was no strong evidence was obtained for fitness influence on the results of the AOA to explain negative differences before and after breakfast. Those results, probably, can be explained by the differences in the oxidative metabolism velocity in the microsomal system of hepatocytes (Cytochrome P450 system) between volunteers, which is genetically determined. But this explanation, of course, needs evidence from large and well-designed trials with strict conditions of meals quantity and composition, simultaneous measurement of blood and skin AOA before and after meals, and specific genetic tests performance.

The data presented in Figure 3 demonstrate the enhancing effect of consumption of food enriched by vitamins on the skin AOA of 6 volunteers (three male and three female) aged 19–34 years. The observed increase in skin AOA after meals enriched with vitamins, which have antioxidative properties (intake of two Askorutin tablets, each containing ascorbic acid (vitamin C) 50 mg and rutoside (flavonoid from vitamin P group) 50 mg) demonstrates, firstly, the informative capabilities of the developed sensory system, and secondly, the direct and rapidly manifested relationship between the internal environment of the body and the skin surface.

## 5. Conclusions

The antioxidant properties of the skin, features of the antioxidant composition of the skin, the role of antioxidants in the physiology of the skin, and role of antioxidant/oxidants in skin health, pathology and interrelation between intrinsic and extrinsic antioxidants known from literature show the importance of antioxidant/oxidants monitoring in skin for health assessment purposes.

Currently, the use of non-invasive methods and sensors of antioxidant activity of skin estimation is limited. This paper contributes to the development in this area. A new disposable sensory system, which is proposed in this work, opens the possibility to investigate large groups of people to evaluate the presence of oxidative stress in skin. Some data illustrating this possibility are presented in this paper. The proposed sensory system can be used in clinical trials, which is important for more in-depth conclusions about the mechanism of the considering processes.

The device and sensory system described in this work, can be applied in on-site and in situ formats.

## Figures and Tables

**Figure 1 sensors-19-02586-f001:**
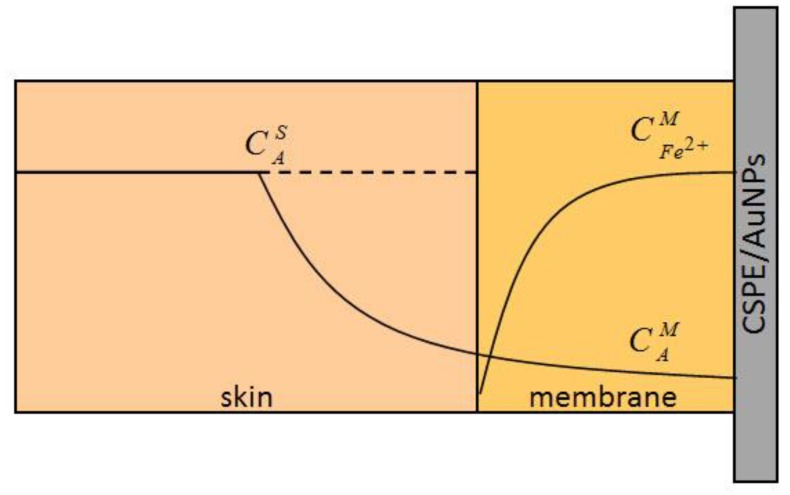
Scheme of delivery of the signal-generating agent (K_4_[Fe(CN)_6_]) to the surface of the indicator electrode (CSPE/AuNPs). $C_{A}^{S}$: the concentration of antioxidants in the skin; $C_{A}^{M}$: the concentration of antioxidants in the membrane; $C_{Fe^{2+}}^{M}$: the increment of concentration of K_4_[Fe(CN)_6_] as a result of reaction (1)in the membrane.

**Figure 2 sensors-19-02586-f002:**
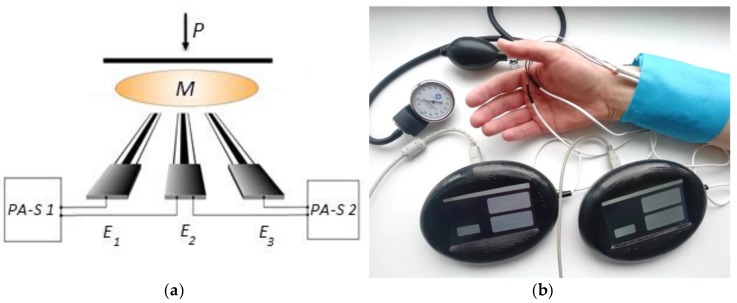
Measuring scheme (**a**) and photo illustrating the process of skin antioxidant activity (AOA) measurement (**b**). E_1_ – E_3_: electrodes; M: membrane; PA-S 1 and PA-S 2: potentiometric analyzers; P: load.

**Figure 3 sensors-19-02586-f003:**
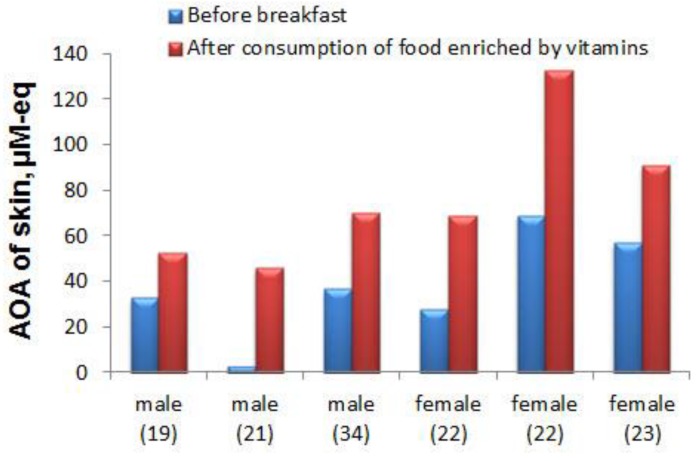
Influence of the use of food enriched with vitamins (100 mg ascorbic acid and 100 mg of rutoside) on the AOA of skin of the respondents. (The age of the respondents is indicated between parentheses).

**Table 1 sensors-19-02586-t001:** Kinetics of stabilization and the values of the potential difference between the same type of electrodes (n = 6, α = 0.05).

Electrode	τ, s	E, mV	R, mV
CSPE	360 ± 55	5 ± 3	7
CSPE/AuNPs	168 ± 72	4 ± 2	5
AgSPE/mod	70 ± 43	1 ± 1	1

τ: potential stabilization time (s); E: steady state potential (mV); R = E_max_ − E_min_ (mV): interval of potential variation.

**Table 2 sensors-19-02586-t002:** AOA of L-ascorbic acid model solutions (n = 4, α = 0.05).

Introduced, μM-eq	Found, μM-eq	RSD, %	Recovery, %
20.0	31.0 ± 16.8	17	155 ± 84
30.0	34.6 ± 3.0	8	115 ± 10
40.0	40.8 ± 2.1	5	102 ± 5
100.0	104.2 ± 2.4	2	104 ± 2
900.0	929.5 ± 22.1	2	103 ± 2

**Table 3 sensors-19-02586-t003:** AOA of skin and L-ascorbic acid recovery (n = 2, α = 0.05).

Respondent No	AOA of Skin, μM-eq	RSD, %	Introduced L-ascorbic Acid, μM-eq	Total AOA, μM-eq	Recovery, %
1	26.0 ± 5.2	20	50.0	81.2 ± 8.8	110 ± 7
2	43.5 ± 6.5	15	50.0	96.0 ± 12.2	105 ± 11
3	64.8 ± 7.8	12	50.0	115.9 ± 9.2	102 ± 3
4	95.4 ± 9.2	9	50.0	145.3 ± 8.0	100 ± 3

**Table 4 sensors-19-02586-t004:** This AOA of skin before (fasting period) and after breakfast (n = 1).

Respondent			AOA, μM-eq		ΔAOA, μM-eq
No	Sex	Age, years	Before Breakfast	After 100 ± 10 min After Breakfast	
1	m a l e	19	8	77	+69
2		18	10	25	+15
3		21	12	40	+28
4		19	73	77	+4
5		18	91	95	+4
6		18	106	73	–33
7	f e m a l e	18	12	69	+57
8		18	22	50	+28
9		22	28	43	+15
10		18	57	40	–17
11		22	69	60	–9
12		23	91	61	–30

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
