# Peer review of "Disposable Potentiometric Sensory System for Skin Antioxidant Activity Evaluation"

_sensors, 2019, doi:10.3390/s19112586_

Reviewer 1 Report

In the reference [28], authors had report the basis and methodology for a non-invasive determination of skin Oxidant/Antioxidant Activity. Now, in this work they propose a disposable contact hybrid potentiometric device for evaluation of skin antioxidant activity, using a modified indicator electrode, a modified reference electrode and a special membrane impregnated by mediator.    

Results showed low values of relative standard deviation (Reproducibility), values of recovery close to 100% and in general a positive difference in AOA evaluated after and before breakfast.

Overall, results of this work have some great potential since results are quite promising, however I would recommend to extend a more detailed explanation of some aspects before it could be considered for publication.

My major comments on this paper are as follows:

1.- In the analysis of the studied population, a more detailed information of the experiment should be reported such as fasting time, kind of meals (quality and quantity), etc.

2.- Discuss what others inclusion criteria could be taken into account to explain negative differences in AOA before and after breakfast

3.- A positive influence of the use of food enriched antioxidants was illustrated in Figure 3, however, statistical differences should be computed and discussed.

4.- What are the arguments considered to argue that “the device can be applied in on-site and in-site formats”

Minor comments:

1.- ROS in line 38 was not defined.

2.- alpha in equation 3 is equal or different to letter "a" in equation 4.

3.- Review whole document for grammatical mistakes.

Author Response

Response to Reviewer 1

1. English language and style – English language and style are fine/minor spell check required (+)

Answer:

The text is edited.

 2. In the analysis of the studied population, a more detailed information of the experiment should be reported such as fasting time, kind of meals (quality and quantity), etc.

Answer:

The information was added: fasting period in the early morning hours after waking up (duration of fasting approximately 1 – 1.5 hours) before breakfast. Regular meal (continental breakfast, without fruits and juices) was taken after the first measurement.

 3. Discuss what others inclusion criteria could be taken into account to explain negative differences in AOA before and after breakfast.

Answer:

The information was added:

Volunteers included into the study have no chronic diseases, their anthropometric data and basic indicators of objective examination (blood pressure, heart and respiratory rate, clinical blood and urine tests, electrocardiography and chest X-ray data) were within normal values, they have no vegetarian or other specific diet.

The regular physical activity (fitness) and special sports nutrition, too, possibly can influence on the AOA of the skin (increase AOA), but the special study is needed to study this relationship. In our study there was no obtained the strong evidence of fitness influence on the results of AOA to explain negative differences before and after breakfast. Those results, probably, can be explained by the differences in the oxidative metabolism velocity in the microsomal system of hepatocytes (Cytochrome P450 system) between volunteers, which is genetically determined. But this explanation, of course, needs the evidence in large and well-designed trials with strict conditions of meals quantity and composition, simultaneous measurement of blood and skin AOA before and after meals and specific genetic tests performance.

4. A positive influence of the use of food enriched antioxidants was illustrated in Figure3, however, statistical differences should be computed and discussed.

Answer:

The task of this work was to create a disposable, easy-to-use, sensory system. The data presented in, obtained only to illustrate its capabilities. Interesting facts shown in fig. 3, we are currently considering as showing a trend. Special medical studies (including statistical analysis) are planned for the future.

 5. What are the arguments considered to argue that “the device can be applied in on-site and in-site formats”

Answer:

 The proof of the validity of this statement is that:

- the sensory system is disposable, its size does not exceed 0.9 × 40 × 0.35 mm;

- the used PA-S device is portable (size 150 × 115 × 55 mm, weight 210 g).

Currently, based on this device / device, we are developing a miniature system with data transfer to the smartphone.

 6. ROS in line 38 was not defined.

Answer:

Explanation of the term is included into the text.

 7. Alpha in equation 3 is equal or different to letter "a" in equation 4.

Answer:

Correction is done.

 8. Review whole document for grammatical mistakes.

Answer:

The text was revised and grammatical mistakes and style were corrected.

Reviewer 2 Report

This manuscript describes a disposable potentiometric sensor based on contact hybrid potentiometric method to measure the skin antioxidant activity. However, there is not information about the sensitivity and the linear rank of the sensor.  Besides, the author should list a table to compare this work with the previous reported sensor in this area.

Author Response

Response to Reviewer 2

1. Is the research design appropriate? – Must be improved (+)

Answer:

It would be fine if respected Reviewer explained what should be improved.

 2. Are the results clearly presented? – Must be improved (+)

Answer:

It would be done if respected Reviewer explained what should be improved.

 3. This manuscript describes a disposable potentiometric sensor based on contact hybrid potentiometric method to measure the skin antioxidant activity. However, there is not information about the sensitivity and the linear rank of the sensor. Besides, the author should list a table to compare this work with the previous reported sensor in this area.

Answer:

In the method used, the linear range is given by the concentrations of the oxidized and reduced forms of the mediator (50 μM K4[Fe(CN)6] / 1000 μM K3[Fe(CN)6]) and is illustrated by the analysis of a model antioxidant (L-ascorbic acid) in concentration range 30–900 μM-eq. These data are given in the Table 2. The sensitivity is determined in the potentiometric method used by the lower limit of the given range. The main difference of the sensory system described in this work from the known ones lies in the system itself: namely, in new electrodes, their manufacture, modification and electrochemical preparation, which allowed it to be used in potentiometric analysis of solutions containing ions interacting with the reference electrode material. Analytical characteristics are determined by the method used and practically do not differ from those previously published, we believe that there is no sense to give it here.

Reviewer 3 Report

This paper systematically investigates a sensing system for skin antioxidant activity detection. The experimental results and system design are interesting. And the practical application are also demonstrated. I think this paper can be published in the current form.

Author Response

Response to reviewer 3

1. English language and style – English language and style are fine/minor spell check required (+)

Answer:

The text is edited.

 2. This paper systematically investigates a sensing system for skin antioxidant activity detection. The experimental results and system design are interesting. And the practical applications are also demonstrated. I think this paper can be published in the current form.

Answer:

Thank you very much!

Round  2

Reviewer 2 Report

accepted